# Symmetric Engineered High Polarization-Insensitive Double Negative Metamaterial Reflector for Gain and Directivity Enhancement of Sub-6 GHz 5G Antenna

**DOI:** 10.3390/ma15165676

**Published:** 2022-08-18

**Authors:** Md. Mhedi Hasan, Mohammad Tariqul Islam, Md. Moniruzzaman, Mohamed S. Soliman, Ahmed S. Alshammari, Iman I. M. Abu Sulayman, Md. Samsuzzaman, Md. Shabiul Islam

**Affiliations:** 1Department of Electrical, Electronic and Systems Engineering, Faculty of Engineering and Built Environment, Universiti Kebangsaan Malaysia, Bangi 43600, Selangor, Malaysia; 2Electrical Engineering Department, College of Engineering, University of Ha’il, Ha’il 81481, Saudi Arabia; 3Department of Electrical Engineering, College of Engineering, Taif University, P.O. Box 11099, Taif 21944, Saudi Arabia; 4Department of Computer and Communication Engineering, Faculty of Computer Science and Engineering, Patuakhali Science and Technology University, Patuakhali 8602, Bangladesh; 5Faculty of Engineering (FOE), Multimedia University, Persiaran Multimedia, Cyberjaya 63100, Selangor, Malaysia

**Keywords:** metamaterial, high polarization-insensitive, symmetric structure, double negative, reflector, gain enhancement

## Abstract

A symmetric engineered high polarization-insensitive double negative (DNG) metamaterial (MM) reflector with frequency tunable features for fifth-generation (5G) antenna gain and directivity enhancement is proposed in this paper. Four identical unique quartiles connected by a metal strip are introduced in this symmetric resonator that substantially tunes the resonance frequency. The proposed design is distinguished by its unique symmetric architecture, high polarization insensitivity, DNG, and frequency tunable features while retaining a high effective medium ratio (EMR). Moreover, the suggested patch offers excellent reflectance in the antenna system for enhancing the antenna gain and directivity. The MM is designed on a Rogers RO3010 low loss substrate, covering the 5G sub-6GHz band with near-zero permeability and refractive index. The performance of the proposed MM is investigated using Computer Simulation Technology (CST), Advanced Design Software (ADS), and measurements. Furthermore, polarization insensitivity is investigated up to 180° angles of incidence, confirming the identical response. The 4 × 4 array of the MM has been utilized on the backside of the 5G antenna as a reflector, generating additional resonances that enhance the antenna gain and directivity by 1.5 and 1.84 dBi, respectively. Thus, the proposed prototype outperforms recent relevant studies, demonstrating its suitability for enhancing antenna gain and directivity in the 5G network.

## 1. Introduction

Metamaterials are unusual engineered composite structure materials that exhibit some stunning electromagnetic characteristics. Negative permittivity, permeability, and refractive index are some attainable properties across a specified frequency range that are not noticed in the conventional materials. These extraordinary properties of the materials were first coined in 1968 by two scientists, V.S. Veselago and John Pendry, who enthralled the globe by arousing insatiable interest in metamaterials [1]. Smith et al. devised the first artificially built and experimentally proved unique metamaterial with effective negative permittivity and permeability, which was made using SRR and thin wire sub-wavelength structures [2,3]. The metamaterial (MM) can be classified as single negative (SNG) or double negative (DNG) based on the remarkable negative properties of permittivity and permeability. Single negative (SNG) metamaterials have either negative permittivity or negative permeability, while double negative (DNG) metamaterials have both negative permittivity and permeability [4]. Negative index, back wave propagation, cloaking, negative refraction, perfect lens, and reverse Doppler shift are the exceptional properties of the DNG MMs that do not comply with the existing fundamental laws [5]. Currently, symmetric and asymmetric shaped metamaterials with stunning DNG properties have been employed in the antenna gain and directivity enhancement for future wireless communications [6,7,8], electromagnetic radiation reduction [9,10], absorber [11] and sensor design [12], wireless power transfer [13], optics [14], cloak and microwave devices [15]. The word ‘metasurface’ refers to a periodic array of unit cells spaced at a specific distance apart that manipulate electromagnetic (EM) waves to improve antenna performance [16]. Metasurface has recently made a breakthrough in optics by surpassing the constraints of conventional optics [14]. Moreover, 5G technology is the next-generation wireless connectivity solution that provides ultra-high-speed and the most secure wireless networks for billions of connected devices while also introducing emerging services such as smart cities, virtual reality, eHealth, IoT, and smart cars. It uses the sub-6 GHz and mm-wave frequency ranges to provide higher data rates with zero latency user experience as well as less interference [17]. In 5G, atmospheric attenuations affect the antenna performance [18], thus demanding the gain enhancement of the antenna to overcome these constraints [19]. In addition, conventional antennas without MM reflectors offer lower gain due to in-phage reflection and surface wave mitigation deficiencies [20,21]. An antenna with an MM reflector can enhance the antenna gain and directivity by reflecting the backside propagation of the antenna and introducing extra resonances in an in-phase fashion. The objective of this study is to improve the gain of a 5G antenna by incorporating an MM reflector.

A flower leaf-shaped DNG MM with frequency reconfigurable features is presented in [22], where the transition frequency is shifted in terms of the number of leaves. A. Hoque et al. recommended a SOCT-shaped compact DNG metamaterial reflector for low-cost liquid salinity sensing applications in the microwave regimes with reflection ability above 90% [12]. The work in [11] proposed a novel spiral-shaped DNG MM for dual-band application with better absorption ability. A symmetrical helix-based DNG metamaterial envelop with metal was analyzed in [23] for backward wave oscillators and amplifiers. This paper improved the device efficiency by fine-tuning the geometry across various bands. In the absence of applications, an asymmetric DNG metamaterial resonator with negative refractive index properties was explored and examined in [24], yet it exhibits a significant EMR. The authors in [25] demonstrate a DNG chiral-shaped MM for dual-band applications with a modest frequency ratio. A triangular-shaped MM with DNG characteristics has been superstrated on a similarly shaped MIMO antenna for enhancing the antenna bandwidth as well as gain [26]. A bilaterally coupled epsilon negative dual-band metamaterial with near-zero permeability and refractive index was reported in [27] for X and C band communication systems. The resonance frequency is tuned in this article using bilateral connected metal width. The authors of [6] introduced a DNG metasurface-loaded MIMO antenna with four MIMO elements for 5G communication, where MM improves the isolation between multiple antenna elements and gain of the antenna. The MM surface was utilized on the backside of the 5G antenna for improving the antenna gain in [28], where the metasurface acts as a reflector. However, the MM structure is asymmetric, and the unit cell structure has received less attention. Authors describe an inductively controlled axis-symmetric metamaterial without DNG features for multiband antenna gain enhancement applications in another study [29], where MM is used as a front cover of the antenna. Similarly, an SEC-SRR-based ENG MM with a near-zero index has been proposed to enhance antenna gain [30]. However, in this article, an asymmetric shaped MM surface has been employed on top of an existing LPDA antenna with inefficient lower band gain. The paper [31] demonstrated a metasurface-loaded DCS-ME dipole antenna with high gain for Ka-band applications, in which the authors employed three layers of metasurface on the antenna’s top to improve the gain. NZIM metamaterial is utilized to enhance the gain of the patch antenna in [32], where the gain is increased by almost 1 dB at the maximum frequencies. Moreover, a complicated structure mm-wave MM-based antenna with lesser gain enhancement is presented in [33]. In [34], the authors reported a 5G antenna with less gain improvement at lower frequencies and a poor radiation pattern, indicating that having the radiating elements on the back side of the antenna is undesirable.

In this study, a symmetric structure high polarization-insensitive DNG metamaterial reflector is suggested for antenna gain and directivity enhancement in the 5G sub-6 GHz spectrum. This symmetric resonator is split into four identical parts, each comprising a unique modified square split ring. The four equal parts are combined with a square-shaped metal strip in the middle of the resonator structure that significantly tunes the resonance frequency. The designed MM structure is distinguished by its simple unique symmetrical structure, high polarization insensitivity, and DNG properties with a near-zero permeability and refractive index. The DNG values have been noticed in the vicinity of the resonance frequency of S_21_, with negative permittivity of 3.6–5 GHz, 6.15–6.34 GHz, and 6.7–7.24 GHz and permeability of 3.74–4.39 GHz, 6.19–6.26 GHz, and 6.75–6.96 GHz, respectively. Moreover, the developed MM structure offers high polarization insensitivity up to 180° and a high EMR value of 8.2, which indicates the design’s compactness. The suggested MM equivalent circuit is developed in ADS, and its performance is confirmed against the findings of the CST simulation. The 4 × 4 array of the MM resonator is employed on the backside of the designed 5G microstrip antenna as a reflector, which gives greater impedance and additional resonances in an in-phase fashion that improves the antenna gain and directivity. In contrast to the antenna without an MM reflector, the antenna with an MM reflector improves gain and directivity by 1.5 dBi and 1.84 dBi, respectively. The 3D far-field radiation characteristics of the MM reflector-loaded and -unloaded antenna have been investigated to clarify the effects of the MM reflector. Thus, our proposed high polarization-insensitive DNG symmetric MM can be utilized to enhance the antenna gain and directivity in the 5G sub-6 GHz wireless network. The manuscript is organized as follows. The second section covers the unit cell design mechanism and simulation approach. The third section illustrates the simulation result, and a detailed discussion of the MM structure, as well as the equivalent circuit of the proposed unit cell, is given in Section 4. Section 5 describes a concise investigation of the surface current density and magnetic and electric field pattern, while Section 6 and Section 7 investigate the resonator’s polarization insensitivity and parametric study, respectively. The array of proposed MM numerical findings is cross-checked against the measured outcomes in Section 8. In Section 9, the proposed MM surface is integrated with the antenna, and its contribution is examined and confirmed by experimental data. The performance comparison table with related recent papers is presented in Section 10, and the paper is concluded in Section 11.

## 2. MM Design Methodology and Simulation Technique

The design mechanism is conducted by the MM sub-wavelength benchmarks in order to achieve the most efficient MM response [35]. Thus, the target of this design is to cover the S and C bands with a symmetric geometric structure, where S is the smallest frequency band that extends from 2 GHz to 4 GHz. Therefore, the initial size of the proposed MM is estimated at 3.3 GHz, which covers the 5G sub-6 GHz spectrum. The preferred unit cell length (*L*) is 10 mm ($L=\lambda_{L}/9.1$, where $\lambda_{L}\;$ is 91 mm at 3.3 GHz), which meets the MM sub-wavelength criterion. The proposed MM is designed based on a standard split ring resonator with an amended composition where four identical subdivided quartiles are connected with a metal strip. Moreover, the diagonal corner at each part is extended by adding a square ring which increases the electrical length of the resonator. The symmetric architecture is achieved by fine-tuning the connecting metal strip, split gaps, and split positions of the resonator. The proposed MM structure is designed on Rogers RO3010 low loss substrate media with an electrical surface area of 0.12λ × 0.12λ × 0.015λ at 3.6 GHz. The permittivity, permeability, and loss tangent of the substrate are 10.2, 1, and 0.0022, respectively. The resonator comprises this dielectric substrate with a copper lossy metal coat, having 0.035 mm thickness. The two rectangular rings are split in the middle of the ring, resulting in four square-shaped equal parts. All similarly shaped split parts are interconnected with the square-shaped metal connector to optimize the MM resonance frequencies. The dimension of this metal strip at the center of the resonator is crucial for tuning the resonance frequency. Each split width and position of the four quartiles is chosen by trial and error, which modifies the operating frequency of the resonator. Two square-shaped metal strips are applied at the two opposite diagonal corners of each part of the resonating patch to enhance the surface of the resonator. All split gaps and structures are equal in length; thus, the suggested MM structure looks symmetric. The developed unit cell design specifications are shown in Figure 1a. The various dimensions of the distinct sections and width of the split gaps and connecting metals are recorded in Table 1. The designed unit cell simulation is accomplished by commercially available numerical analysis microwave simulation software CST studio suite, 2019, in the frequency domain-based solver. The unit cell simulation setup is depicted in Figure 1b, where the electric field and magnetic field are applied on the X-axis and Y-axis, respectively. The electromagnetic wave, on the other hand, excites in the Z-axis direction, which is properly spaced from the reference plane.

## 3. Unit Cell Performance

The developed unit cell simulation result is examined by the CST studio suite, 2019, with a robust retrieval model-based built-in post-processing method. The transmission and reflection coefficients spectra of the proposed MM structure are presented in Figure 2, which covers the dual bands of *S*_21_ at three resonances, 3.6 GHz, 6.15 GHz, and 6.71 GHz, consecutively. The electrical resonance is confirmed in the *S*_11_, as each resonance of *S*_11_ appears following the resonance of *S*_21_ indicated in Figure 2. The effective permittivity, permeability, impedance, and refractive index are determined from the built-in post-processing technique in the CST electromagnetic simulator. A robust retrieval method is used to derive the MM effective parameters from the scattering parameters. The refractive index and impedance can be derived using the reflection and transmission coefficient Equations (1) and (2) (see reference [36]).

Reflection coefficient:(1)$$S_{11}=\frac{R_{01}\left(1-e^{i2nk_{0}d}\right)}{1-R^{2}{}_{01}e^{i2nk_{0}d}}$$

Transmission coefficient:(2)$$S_{21}=\frac{(1-R^{2}{}_{01})\;e^{ink_{0}d}}{1-R^{2}{}_{01}e^{i2nk_{0}d}}$$
where $R_{01}=\frac{z-1}{z+1}$.

The impedance and refractive index are calculated by reversing Equations (1) and (2):

Impedance:(3)$$z=\pm \sqrt{\frac{{\left(1+S_{11}\right)}^{2}-S_{21}{}^{2}}{{\left(1-S_{11}\right)}^{2}-S_{21}{}^{2}}}$$

Refractive index:(4)$$n=\frac{1}{K_{0}d}\;\left\{\left[{\left[\ln\left(e^{ink_{0}d}\right)\right]}^{\prime \prime }+2m\pi \right]-i\;{\left[\ln\left(e^{ink_{0}d}\right)\right]}^{\prime }\right\}$$
where the real and imaginary elements of the operator are denoted by (.)’ and (.)”, respectively, while the integer number *m* correlates to the real refractive index. The permittivity and permeability are calculated using the equations ε=n/z and μ=nz, respectively, which are based on the impedance and refractive index Equations (3) and (4).

The effective retrieve permittivity of the unit cell depicted in Figure 3a indicates the negative permittivity at 3.6 GHz, 6.17 GHz, and 6.72 GHz, respectively, which are very close to the *S*_21_ resonances. The effective permeability of the MM structure is illustrated in Figure 3b, which demonstrates the negative permeability for all resonance frequencies of *S*_21_. It is noteworthy that the real effective negative permeability region expands from 3.74 to 4.39 GHz, 6.19 to 6.26 GHz, and 6.75 to 6.96 GHz, respectively, which follow each of the resonance frequencies of *S*_21_ within the negative permittivity region. Moreover, the resonances of the negative permeability are synchronized tightly in the maximum negative regions of the effective permittivity. Hence, the proposed MM structure can be deemed the Double Negative (DNG) MM. The retrieve and effective refractive index of the proposed unit cell is depicted in Figure 4a alongside the frequency. It is noticed that the proposed MM confirms a near-zero refractive index at 3.6 GHz, 6.17 GHz, and 6.72 GHz, consecutively, within the spectra of *S*_21_. Moreover, the frequency regimes of the effective near-zero index exist within the permittivity and permeability frequency regimes. Table 2 represents the extracted distinct properties of the effective parameters of the developed DNG unit cell structure. The real and imaginary effective impedance values of the proposed MM are presented in Figure 4b, where the effective real impedance is positive according to the *S*_21_ resonances; thus, the proposed unit cell is a passive media. The negative imaginary values are observed to be near the *S*_21_ resonances. The impedance of the MM is strongly related to its effective permittivity and permeability; therefore, by modulating the permittivity and permeability, we can achieve excellent impedance matching with the open space surrounding the MM.

## 4. Equivalent Circuit Analysis of the Proposed Unit Cell

The metamaterial structure contains four equal square-shaped quartiles with a central connecting metal strip that exhibits inductive and capacitive effects. The inductance is induced by the metal strips, whereas split gaps contribute to the capacitance of the structure, which controls the resonance frequency of the resonator. By combining the split and the electric field, it is possible to produce electric resonance. Conversely, when a resonating structure interacts with an electromagnetic wave, magnetic resonance is created by the combination of metal loops and magnetic field response. Thus, the whole structure resembles an LC circuit that exhibits resonance, with resonance frequency (f) having the following relation [37]:(5)$$f=\frac{1}{2\pi \sqrt{LC}}$$
where *L* and *C* are the inductance and capacitance of the resonator, respectively. The capacitance, *C*, owing to the split gap, can be determined by using the relation as expressed in Equation (6):(6)$$C=\varepsilon_{0}\varepsilon_{r}\frac{A}{d}\left(F\right)$$
where $\varepsilon_{0}\;$ and $\;\;\varepsilon_{r}$ denote the free space permittivity and relative permittivity, respectively, and *A* and *d* indicate the metal strip area and split distance, respectively. The inductance of the resonator ring can be computed by applying the transmission line principle equation [38]:(7)$$L\left(nH\right)=2\times 10^{-4}l\left[\ln\left(\frac{l}{w+t}\right)+1.193+0.02235\left(\frac{w+t}{l}\right)\right]K_{g}$$
where the correction factor, $\;K_{g}=0.57-0.145\;\ln\frac{w^{\prime }}{h^{\prime }}$, in which $w^{\prime }$ and $h^{\prime }$ are the substrate width and thickness, respectively. Moreover, t,  l, and w denote the thickness, length, and width of microstrip lines, respectively. The equivalent circuit of the proposed metamaterial unit cell is drawn in Figure 5a, which considers the inductive and capacitive effects of each ring as well as the split gaps. In the proposed MM, four modified square-shaped split rings are connected at the center of the resonator through inductive metal strips. In the designed equivalent circuit, four square rings are represented by inductances *L1*, *L2*, *L3*, and *L4*, whereas corresponding capacitances due to the split gaps are presented by capacitances *C1*, *C2*, *C3*, and *C4*. Since four rings are interconnected by small metal strips, this interconnection can be represented with low impedance interconnecting inductances *L5* and *L6*. The capacitance *C5* is associated with the co-planar capacitances between the rings. Two ports are connected at the two ends of the circuit that is terminated with the 50 Ohm impedances. The S-parameter module in advanced design system (ADS) is used, providing a virtual environment for frequency allocations as well as for electromagnetic response. This equivalent circuit is modeled in ADS software, and the effects of the various inductances and capacitances on the resonance of *S*_21_ are investigated. The simulation in ADS begins with the starting values of 1 nH for each inductor and 1 pF for each capacitor. Then, through the tuning module in ADS, the values of the components are adjusted through numerous simulations so that the transmission coefficient spectrum obtained from the equivalent becomes similar to that obtained from MM unit cell simulation in CST. Thus, component values are obtained for the equivalent that is displayed on the circuit presented in Figure 5a. In this equivalent circuit, *L1* and *C1* component values are adjusted to control the resonance at 3.6 GHz. The resonance frequency is determined by the values of these two components, which are coupled in series to form an LC resonance circuit. Inductance *L2* and *C2* show their impact on the upper cut-off frequency of the resonance at 3.6 GHz. Additionally, these component values are responsible for wave-shaped fluctuation at high frequencies.

The resonance at 6.71 GHz is controlled by the inductor–capacitor pair *L3* and *C3*. A variation in these component values causes a shift in resonance around 6.7 GHz. Finally, *L4* and *C4* assist in obtaining the resonance at 6.15 GHz. The coupling inductance *L5* and *L6* have lower values, as they provide a low impedance, short-circuit path among the branches. A high value of these components significantly suppresses the waves. The effect of variation in these inductor’s values is more prominent in high-frequency resonances, whereas the low frequency is less prone to changes in values of these parameters. The effect of *C5* is observed to determine the wave shapes of the entire frequency range. Moreover, *C5* controls all cut-off frequencies of the discussed three resonances, and the appropriate selection of this component value provides a more similar response to the *S*_21_ response obtained in CST. Thus, through tuning, component values are finalized, and from the circuit simulation, the *S*_21_ response is obtained. Figure 5b reveals a comparison of *S*_21_ responses for CST and ADS. As seen in Figure 5b, the equivalent circuit closely resembles the MM unit cell, as its *S*_21_ response is identical to that of the CST simulation of the unit cell. A mismatching in magnitudes of *S*_21_ is observed in Figure 5b, where circuit simulation in ADS shows higher values of negative magnitude at the resonance frequencies. This mismatching can be eliminated considering the resistive effect associated with metal strips. However, for the circuit simplicity, the resistive effect is ignored in the present circuit modeling. Despite these slight deviations, the equivalent circuit exhibits agreements with the proposed MM unit cell and thus represents the MM unit cell.

## 5. Surface Current Density, Magnetic Field, and Electric Field Studies

The liaison between the current, electric field, and magnetic field distribution in the MM can be described using Maxwell equations, which have a significant effect on the resonance phenomena of the unit cell. Since the magnetic field is produced due to the change in the electric field with time and vice versa, the changing magnetic field provides the electric field with varying times. Therefore, instead of utilizing the traditional field equation, the MM’s effective medium parameter uses Equations (8) and (9) to illustrate the magnetic and electric fields [39].
(8)$$B_{ave}=\mu_{eff}\mu_{0}H_{ave}$$
(9)$$D_{ave}=\varepsilon_{eff}\varepsilon_{0}E_{ave}$$

The flux densities associated with the *E* and *H* fields can be stated using Maxwell’s integral equations:(10)$$\int_{C}^{}H.dI=0+\frac{\partial }{\partial t}\iint_{S}^{}D.dS$$
(11)$$\int_{C}^{}E.dI=0-\frac{\partial }{\partial t}\iint_{S}^{}B.dS$$

The unit cell surface current, electric field, and magnetic field distribution and propagation of the electromagnetic wave are demonstrated by Equations (10) and (11). The suggested unit cell split gaps and connecting metal rings contribute to the inductive and capacitive effects. When the proposed MM resonator is excited by an electromagnetic wave, it senses electromagnetic force. As the suggested MM structure conforms to the subwavelength standards, the induced current moves through the resonator and influences relative permeability and permittivity characteristics. The surface current of the proposed MM structure is portrayed in Figure 6 for three resonances. At 3.6 GHz, it is noticed that the maximum current density is seen on the opposite side of the split gaps of all square rings and the current is almost negligible around the split gaps due to capacitance effects, whereas the current is directed towards the right side parallelly through the connected metal strip. Thus, the magnetic resonance occurs at the point where the response of the magnetic field is higher than the electric field response, shown in Figure 7a and Figure 8a, respectively. At 6.15 GHz, the current is reduced significantly in the resonator, and a few antiparallel currents are noticed in the middle horizontal part of the unit cell, as demonstrated in Figure 6b. As the current is followed antiparallelly, the electric field intensity is stronger than the magnetic field intensity, implying the electric resonance point presented in Figure 7b and Figure 8b. Two strong vertical current loops appear in the inner vertical arms of the resonator, indicated in Figure 6c; the current is also seen in the horizontal middle part of the resonator. At 6.71 GHz frequency, electric and magnetic resonances arise due to the intense magnetic and electric fields, which can be seen in Figure 7c and Figure 8c, respectively. At higher frequencies, the split gaps of the resonator structure conduct negligible currents, as higher frequencies lead to higher impedance. The magnetic field distribution of the suggested unit cell is shown in Figure 7a–c, which is entirely reliant on the surface current density as defined by Amperes law: B=μI/2πr, where *μ* is the permeability constant, and *r* is the distance from the metal wire. It is noteworthy that the magnetic field is stronger for 3.6 GHz and 6.71 GHz, and lesser for 6.15 GHz. Figure 8a–c illustrates the electric field distributions at 3.6 GHz, 6.15 GHz, and 6.71 GHz, respectively, where a strong electric field is observed around the split gaps for certain *S*_21_ resonances due to capacitance effects, although the electric field intensity is lower at 3.6 GHz compared to the two higher resonance frequencies. The capacitance and resonances are produced by the split gaps of the developed MM structure. It is also observed that the excitations of the electric and magnetic fields are opposite each other for resonances, meeting the Maxwell equation requirements.

## 6. Polarization Insensitivity Analysis

As the proposed MM structure is symmetrically engineered, its performance is consistent with various incidence angles of electromagnetic waves. The polarization insensitivity of the proposed DNG MM structure has been investigated using polarization angles (φ) ranging from 0° to 180° and incident angles (θ) ranging from 0° to 90°, as revealed in Figure 9a,b. The floquet boundary conditions were used to investigate the polarization features of the MM. The unit cell boundary conditions have been imposed along the x-axis and y-axis while the z-axis is open. As illustrated in Figure 10, the propagation k’s wave vector is in the z-direction, while the E and H vectors are in the x- and y-directions, respectively. At all incidence angles up to 180°, high polarization insensitivity can be seen in the suggested MM unit cell demonstrated in Figure 9a. Under both normal and oblique incidence of electromagnetic waves, the DNG unit cell exhibits identical resonances (Figure 9a,b). Thus, the developed MM structure is highly insensitive to all polarizing angles because of its unique symmetric design geometry. The proposed MM structure could be a feasible solution for 5G applications since the unit cell response is consistent across all angles of incident waves.

## 7. EMR and Parametric Study

The unit cell dimension compactness and acceptability are the crucial issue nowadays for applications in 5G wireless communication. The numerical approach for determining the compactness of the MM unit cell is the effective medium ratio (EMR). The following equation has been used to determine the EMR of the developed MM structure:$\;EMR=\lambda_{o}/L$, where $\lambda_{o}$ denotes the wavelength at the lower resonance frequency, and *L* denotes the size of the unit cell [29]. Thus, the EMR value of the proposed MM structure is 8.2 for $\lambda_{o}=82$ mm and L=10 mm, which confirms a compact size of the developed MM structure.

The split gaps in the rings act as an LC circuit, which generates the resonances according to the value of the L and C. Thus, the capacitance of the split gaps and the inductance of the rings have a significant impact on the resonances of the proposed MM structure. The width of the split gaps and the size of the connecting metal strip in the middle of the unit cell have been studied, these impacts on the resonance frequencies of the MM structure have been observed. The scattering parameters of the proposed unit cell are presented in Figure 11a,b for various split gaps and middle connector metal sizes, respectively. In Figure 11a, it is observed that the various split gaps *‘g’* exhibits a substantial effect on the higher resonances of the *S*_21_ than the lower ones. The frequency is dramatically shifted towards the lower frequency when the split gaps are decreased from 0.9 mm to 0.5 mm, making it possible to tune the resonance frequency by adjusting the split gaps of the resonator. The numerous sizes of the middle connecting metal strip *‘d’* have been analyzed to determine the impact on the resonances of the *S*_21_, as displayed in Figure 11b. The analysis demonstrates no significant effects on the lower two resonance frequencies; however, there are considerable variations in the resonance frequency at 6.71 GHz. Furthermore, the number of resonances is controlled by the various sizes of the connecting metal strip, as indicated in Figure 11b, which is a unique characteristic of this suggested MM resonator. At 6.71 GHz, the resonances are modulated by about 24%, as shown in Figure 11b, with a variation in the size of the interconnecting metal strips indicating frequency hopping characteristics. In addition, for *d* = 3.5 mm, the resonance frequencies are observed at 3.75 GHz, 6.18 GHz, and 8.3 GHz, which covers the S, C, and X bands, respectively, whereas the resonance frequencies for other values of ‘*d*’ offer only two bands, implying that the frequency band rises with the various connecting metal values. Thus, for 5G wireless communication applications, it is feasible to tune the resonance frequency of the proposed resonator by varying the width of the split gaps and connecting metal strips.

## 8. Experimental Results and Analysis

The experimental arrangement of the developed DNG metamaterial is presented in Figure 12a, which includes a fabricated prototype photograph of the proposed 4 × 4 unit cell array captured at microwave lab, UKM, Malaysia. In the measurement setup, an Agilent PNA series network analyzer is utilized, coupled with distinct waveguide ports for transmitting and receiving signals. The 4 × 4 array prototype is positioned between the two waveguide ports that are connected to the two-port network analyzer (Agilent PNA N5227A) through the coaxial cable. The scattering parameters of the fabricated suggested prototype were observed, employing two pairs of distinct waveguide ports (A-INFOMW, P/N:187WCAS and A-INFOMW, P/N:137WCAS). The network analyzer in the experimental setup was calibrated using an Agilent N4694-60001 calibration kit. Figure 12b compares the CST simulation outcomes with the measured transmission coefficient results of the developed array prototype. It is noted that the outcome of the simulated array is nearly identical to that of the simulated unit cell, with a few slight deviations. These disparities are triggered by a mutual coupling effect between the array components. The experimental resonance frequencies of the *S*_21_ are found to be quite identical to the simulated one, as illustrated in Figure 12b. Furthermore, the measured transmission coefficient of the array prototype encompasses the 5G sub-6 GHz spectrum, implying that 5G applications are conceivable, although the measured resonance frequencies, bandwidth, and amplitude deviated slightly from the CST-simulated findings. Fabrication defects, measurement tolerance, open-air measurement arrangement, size mismatch between waveguide port and fabricated prototype, and impacts of mutual coupling between waveguide ports are all factors that contribute to the discrepancies between measured and simulated results. Moreover, the right placement of the fabricated prototype between waveguide ports in the experimental arrangement contributes to the shifting of resonances. In addition, some undesirable noises were observed during the calibration phase (for both distinct waveguide ports), contributing to the discrepancy between numerical and measured results. However, despite these shortcomings, the array performs well, with accordance between simulation and measurement, making it suitable for 5G wireless applications.

In order to verify the polarization insensitivity performance of the proposed MM, the experimental investigation of the suggested symmetric structure high polarization-insensitive MM is performed. Figure 13 displays a photograph of the far-field measurement system with the fabricated prototype (200 mm × 200 mm dimension, comprising 20 × 20-unit cells), in which the suggested prototype is positioned between two horn antennas. The two horn antennas are connected with a two-port VNA (Agilent N5227A) to measure the *S*_21_ and *S*_11_ for normal and oblique incidence. The scattering parameters of the proposed prototype were measured by rotating the horn antenna for different polarization and incident angles. Figure 14a,b shows the polarization insensitivity experimental results for φ = 0° to 180°, and Figure 15a,b reveals the oblique angle insensitivity experimental results up to 90° angles of incidence. The measured scattering parameters for normal and oblique incidence are mostly consistent with simulation findings, as demonstrated in Figure 14a,b and Figure 15a,b. It is observed that the slight divergence happened in the experimental and numerical results owing to the mutual coupling effect between unit cells, fabrication defect, and measurement tolerance. Moreover, the precise placement and alignment of the fabricated prototype between horn antennas in the experimental arrangement contribute to the shifting of resonances. In addition, some undesirable noises were observed during the calibration phase, contributing to the discrepancy between numerical and measured results. Despite these influences, the proposed MM performs well, with accordance between simulation and measurement, confirming the high polarization insensitivity owing to the symmetric resonator structure.

## 9. 5G Antenna with MM Reflector

The proposed symmetric MM structure with DNG and near-zero characteristics has considerable potential in 5G and beyond wireless communications for improving antenna gain [28,40] and directivity [41]. A simple 5G slotted monopole microstrip patch antenna with a defected ground plane has been devised to examine the suggested MM performance in terms of boosting the antenna gain and directivity. The developed slotted antenna operating at 3.5 GHz has been printed on the low-cost FR-4 substrate with a thickness of 1.6 mm, which is demonstrated in Figure 16a. The rectangular patch of the designed antenna has been slotted twice; first, the patch’s four corners have been slotted with a square ring, and subsequently, the patch’s left, right, and upper center have been slotted with a circular ring. The final optimized desired design parameters of the developed antenna are as follows: *S_L_* = 30 mm, *S_w_* = 35 mm, *P_w_* = 26 mm, *P_L_* = 20 mm, *r* = 3.25 mm, *s* = 3 mm, *f_L_* = 8 mm, *f_W_* = 2.64 mm, *f_1_, f_2_* = 2 mm, *f_3_* = 10 mm, *f_4_* = 25 mm, and *f_5_* = 10.5 mm. The reflection coefficient (S_11_), voltage standing wave ratio (VSWR), and gain of the developed antenna (without MM) are presented in Figure 16c and Figure 17a,b, respectively. It is seen from Figure 16c that the antenna is operating at 3.5 GHz (frequency range: 3.4 GHz to 3.6 GHz, 5G sub-6 GHz band) with a −29.1 dB magnitude. The antenna exhibits a low gain of 3.21 dB, which is not suitable for 5G applications, as shown in Figure 17b. Thus, to enhance the antenna gain, the developed antenna is positioned ahead of the MM reflector. The antenna dimension is 30 mm × 35 mm, so 4 × 4 arrays of MM unit cells with a dimension of 40 mm × 40 mm have been used at the height of 11.5 mm, as depicted in Figure 16b. The backward metasurface acts as a reflector, which offers a resonant cavity that produces supplemental resonances, thus improving the antenna radiation efficiency and enhancing the antenna gain. The realized gain and directivity of the antenna are enhanced when the reflector-generated wave is in phase with the antenna radiation direction. The gap between the metasurface and the antenna is crucial for achieving high gain with productive interference between the antenna-induced wave and the metasurface-generated wave. Several simulations are utilized to optimize the height between the antenna and the MM reflector, as well as the position of the MM reflector, in order to achieve high gain and directivity. The reflection coefficient (S_11_) curve of the antenna with an MM reflector is nearly identical to that of the antenna without an MM reflector, as illustrated in Figure 16c. The slight variation is caused by the metasurface reflected wave, which influences the antenna’s surface current. The voltage standing wave ratio (VSWR) is also investigated to validate the slight variation in the S_11_ of the developed antenna, as shown in Figure 17a. The Voltage Standing Wave Ratio (VSWR) is a performance indicator that determines how well power is transferred to the load, i.e., how much reflection loss there is, which is defined by $VSWR=\frac{1+\left|S_{11}\right|}{1-\left|S_{11}\right|}$. If the VSWR is less than 2, the antenna reflects less power, suggesting that it is properly matched; however, if it is more than 2, information loss occurs due to high reflecting power and poor impedance matching, necessitating the installation of an external impedance circuit. The values of VSWR of the antenna without and with an MM reflector are 1.09 and 1.1, respectively, which ensure the antenna is well matched and does not require supplementary impedance matching. The gain of the antenna without and with an MM reflector is observed at 3.21 dBi and 4.69 dBi, respectively. However, the antenna gain with an MM reflector is enhanced significantly by 1.5 dBi, or around 46%, compared to without an MM reflector, as indicated in Figure 17b. Furthermore, we have investigated the antenna gain performance by employing a plane copper-reflecting surface on the backside of the suggested antenna with a height of *h*. The gain performance of the plane copper-reflecting surface in the antenna system is depicted in Figure 18. It is observed that the plane copper surface decreases the antenna gain when *h* is 11.5 mm. At the height of 27 mm, the maximum gain enhancement of 0.77 dBi is observed, which increases the antenna dimension and limits the application area. Thus, the suggested MM might be a promising choice for 5G sub-6 GHz antenna gain enhancement.

The 3D radiation patterns of the antenna loaded without and with a metasurface reflector are presented in Figure 19a,b, respectively. As demonstrated in Figure 19a,b, the directivity of the MM-loaded antenna improved by around 33% over that of the unloaded antenna. The MM array reflector reduces the antenna’s back loop and produces extra radiation that directs currents in the z-direction (Figure 19b), resulting in enhanced gain and directivity. Thus, our suggested highly polarization-insensitive symmetric DNG MM structure is useful for boosting the gain and directivity of the antenna system in 5G wireless communications.

**Experimental Analysis:** The performance of the proposed 5G antenna with and without the DNG reflector, notably gain enhancement, has been evaluated and validated by experimental data. In Figure 20a, a photograph of the fabricated proposed 5G antenna and 4 × 4 array of the DNG unit cell with assembled components is illustrated. The single 5G antenna is associated with an SMA connector, after which the suggested DNG MM is positioned on the rear side of the antenna at a 11.5 mm distance using the polystyrene slab as a spacer. The fabricated prototype is connected to the two-port network analyzer (Agilent PNA N5227A) to collect experimental S-parameter data. Figure 20b shows the simulated and experimental *S*_11_ plots with and without the MM reflector. It is indeed noticeable that the proposed 5G antenna system with and without an MM reflector operates at the 5G sub-6 GHz spectrum, as demonstrated by simulations and laboratory experiments. Although slight variations in the degree of amplitude and operating range exist between measured and simulated outcomes, the 5G sub-6 GHz frequency spectrum has been effectively covered. Fabrication flaws, coaxial and SMA connection loss, and instrument calibration inaccuracies all contributed to these obstacles. Furthermore, the antenna and MM reflector are sandwiched by a polystyrene-based spacer, which is another concern that influences the observed findings compared to the predicted results.

Figure 21a illustrates the measured VSWR values for the proposed antenna with and without MM, which is consistent with the simulated ones, with a value of less than 2, indicating that the antenna meets the low power loss threshold. Figure 21b illustrates the framework of the SATIMO nearfield experimental process used to examine gain, efficiency, and radiation pattern at UKM’s SATIMO near-field system lab. The total efficiency of the single antenna and antenna with an MM reflector is presented in Figure 22a. The observed maximum efficiency is approximately 55% for the non-MM antenna and 62% for the MM reflector antenna at the resonance frequency, with minor variance noted compared to the simulated values. This could happen as a consequence of measurement equipment imperfections as well as the spacer that is employed between the antenna and the MM reflector, among other reasons. As measured findings, the suggested MM reflector enhanced overall efficiency by 7%, indicating MM reflector effects. The gain curve of the antenna is presented in Figure 22b, which incorporates measurement and simulation results for both with and without an MM reflector. It is noted that the gain is enhanced by the MM reflector, which is fitted behind the antenna, and this is clearly verified by the experimental results. The simulated and experimental results are nearly identical; however, at some frequencies, notably those in the upper band, the measured gain is greater than the simulated gain when the antenna is encountered by the MM reflector. Nevertheless, the measured gain drops at lower frequencies, although this is not a significant concern because the highest viable 5G sub-6 GHz frequency range (3.3–3.8 GHz) gives outstanding gain. This fluctuation in experimental gain is driven by the coupling polystyrene spacer in the antenna system. The measured gain at resonance frequency is nearly similar to the simulated gain, showing that the proposed MM reflector performs as expected and that the recommended reflector-based antenna is suitable for 5G applications. Owing to the error involved in the experimental investigation, the overall laboratory experimental results differ from the simulated findings. The proposed prototype performance is influenced by impedance mismatch between the antenna and the SMA connector, coaxial cable loss, room temperature, soldering effect, and the proximity of various electronic equipment to the experimental setup.

**Radiation pattern:** The simulated and measured 2D radiation patterns of the suggested antenna without and with an MM reflector in the E-plane (co- and cross-polarization) and H-plane (co- and cross-polarization) at 3.5 GHz are demonstrated in Figure 23a,b, respectively. The antenna with an MM reflector offers stable and unidirectional radiation patterns by reducing the back-lobe and side-lobe radiation at the operating frequency. It is observed that the experimental radiation pattern is almost identical to the CST-simulated radiation pattern; a slight variation is due to the misalignment of the various parts of the assembly, fabrication defect, measurement tolerance, terminated port reflection, the spacer between antenna and MM reflector, and cable connection loss.

## 10. Performance Comparison with the Existing Contributions

The proposed polarization-insensitive symmetric DNG MM performance is compared to recently published literary works in the relevant area, as shown in Table 3. The comparisons were conducted based on structural type, negative characteristics, operational frequency ranges, EMR, polarization insensitivity, and antenna gain enhancement.
materials-15-05676-t003_Table 3Table 3Contrast between the proposed MM and existing relevant work.Ref. and YearUnit Cell Size (mm^2^) and Shape ENG/DNGFrequency BandsEMRPolarization Insensitivity (Maximum Angles)Gain Enhancement (dBi)Applications[42]20159 × 9AsymmetricNot shownKuNot shownNot shown1.0Radar cross-section (RCS) of reduction[43]201716 × 16Axis SymmetricDNGX1.7540°Not shownMicrowave absorber(proposed)[44]20175 × 4.8SymmetricNot shownKuNot shownNot shown1.15Gain enhancement[45]20189.25 × 9.25SymmetricNot shownKuNot shown90°-Antenna performance improvement[25]20187.7 × 7.7SymmetricDNGC, X5.9Not shownNot shownMicrowave (proposed)[11]201910 × 10AsymmetricDNGKu, K280°Not shownMicrowave absorber(proposed)[46]20196 × 6AsymmetricNot shownCNot shownNot shown1.3Gain enhancement[28]202011 × 11AsymmetricENGSNot givenNot shown3.5Gain enhancement[47]20218 × 8SymmetricLHMX, Ku4.32745°Not shownCommunication (Proposed)[9]202120 × 13AsymmetricMNGX1.5Not shown0.3Gain enhancement[12]202115 × 15AsymmetricDNGC3.5Not shownNot shownMicrowave sensing[48]202210 × 10Axis SymmetricENGCNot shownNot shown0.6Gain enhancement**Proposed****10 × 10****Symmetric****DNG****5G Sub-6 GHz****8.2****180°****1.5****Antenna gain enhancement**

The studies in [9,11,12,28,42,46] presented a variety of low-EMR MMs based on asymmetric resonator structures without frequency adjustable features, whereas the suggested symmetric engineered DNG MM exhibits high EMR and frequency tunable qualities with near-zero permeability and refractive index. Moreover, the developed MM shows high polarization-insensitive characteristics up to 180° angle of incidence and antenna gain enhancement properties in the 5G sub-6 GHz spectrum. However, the MMs in [25,43,44,45,47,48] have a symmetric resonator with DNG characteristics, but their EMR values are relatively low. Additionally, the authors in [11,25,43,47] demonstrate low EMR values without considering any applications compared to our proposed work. On the other side, the MMs in [11,43,45,47] exhibit polarization insensitivity up to maximum 90° polarization angles, whereas our proposed MM offers high polarization insensitivity up to 180° angles of incidence. Gain enhancement has been demonstrated in [9,42,44,45,46,48] with ENG characteristics, but the proposed antenna system has a higher gain enhancement than the reported prototype. Although the authors of [28] achieved a higher gain enhancement than the suggested prototype, the air gap between the MM reflector and antenna is higher (more than 30 mm), which enhances dimension. Aside from that, the MM and antenna dimensions are larger than the proposed prototype, which raises the cost and limits the application area. Furthermore, the MM reflector reduced impedance bandwidth while shifting the antenna resonance frequency. Thus, our proposed high polarization-insensitive symmetric engineered DNG metamaterial can be utilized to improve antenna gain and directivity in 5G sub-6 GHz wireless communication.

## 11. Conclusions

A high polarization-insensitive symmetrical structured DNG metamaterial reflector with frequency tunable properties for 5G antenna gain and directivity enhancement applications is presented in this manuscript. The proposed symmetric engineered unique resonator was realized on a Rogers RO3010 low loss substrate with an electrical dimension of 0.12λ×0.12λ at 3.6 GHz. Negative permittivity, permeability, and near-zero refractive indexes were close to the *S*_21_ resonances, occurring at 3.6 GHz, 6.15 GHz, and 6.71 GHz, respectively. The presented unit cell performed as a DNG metamaterial with a high EMR value of 8.2, indicating its appropriateness and compactness. Furthermore, it was observed that the developed MM structure is polarization-insensitive up to 180° incidence angle, which is unique among relevant metamaterials to date. The surface current density and the electric and magnetic field distributions were studied to explore the metamaterial characteristics. The MM equivalent circuit was modeled and validated using the ADS by comparing its *S*_21_ with the simulation result of CST. The performance of the MM array was examined by measurements and confirmed using CST simulations. The proposed MM surface was employed on the backside of the 5G antenna as a reflector to improve its gain and directivity. Numerous simulations were performed to optimize the air gap between antenna and reflector. The MM reflector notably enhanced the antenna gain and directivity by 1.5 dBi and 1.84 dBi, respectively. The high impedance metasurface reflector offers additional resonances, which improves antenna gain and directivity while providing a stable radiation pattern with low back-lobe and side-lobe levels. The experimental and simulation findings were found to be quite close to each other. Owing to its unique symmetric architecture with DNG and high polarization insensitivity, near-zero permeability, and refractive index, it is well suited for applications in 5G sub-6 GHz wireless communication, notably for antenna gain and directivity enhancement.

## Figures and Tables

**Figure 1 materials-15-05676-f001:**
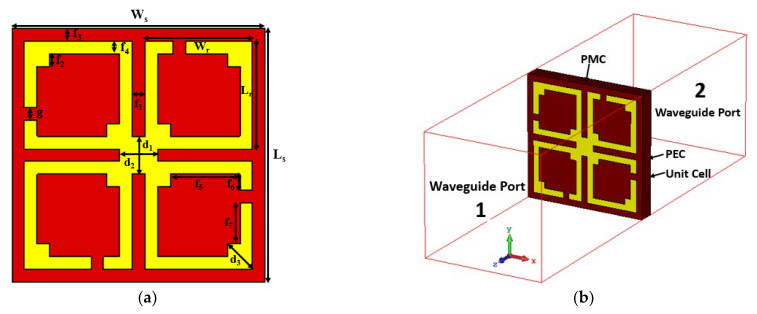
(**a**) Double negative (DNG) metamaterial with a symmetric structure and (**b**) unit cell simulation with boundary setup.

**Figure 2 materials-15-05676-f002:**
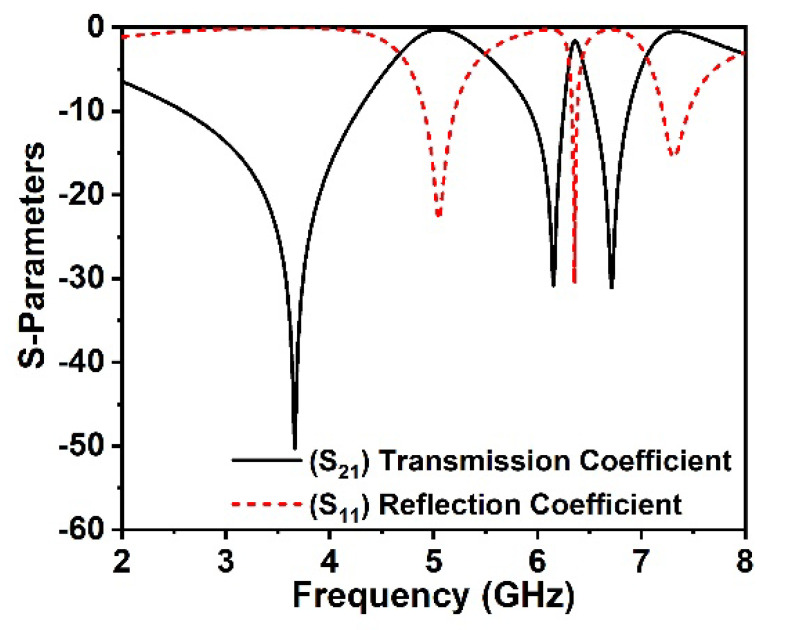
Reflection and transmission coefficients of the proposed MM structure.

**Figure 3 materials-15-05676-f003:**
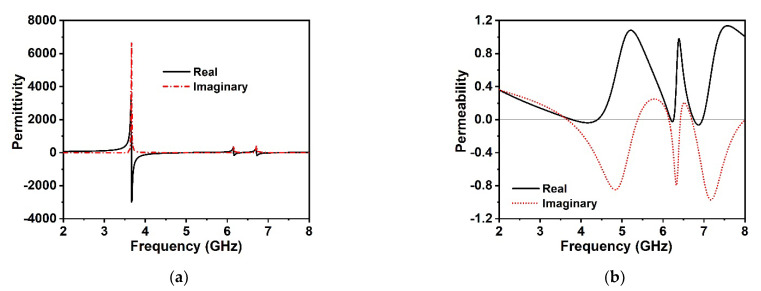
(**a**) Effective permittivity values and (**b**) effective permeability values with the DNG region of the proposed MM.

**Figure 4 materials-15-05676-f004:**
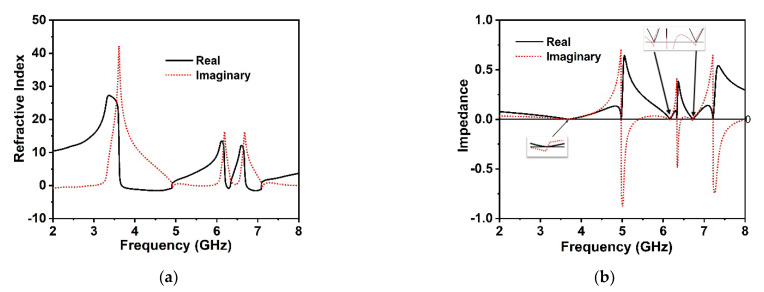
(**a**) Refractive index quantities and (**b**) effective impedance against the frequency of the proposed MM structure.

**Figure 5 materials-15-05676-f005:**
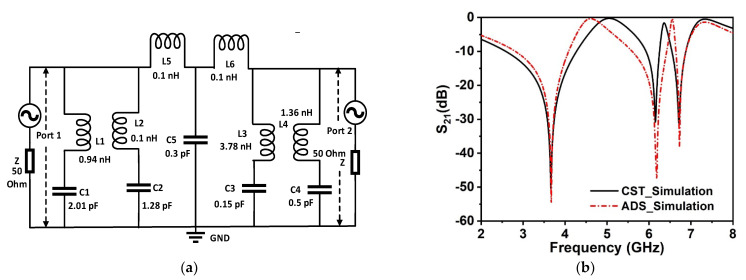
(**a**) Equivalent circuit of the proposed MM unit cell. (**b**) Transmission coefficient (*S*_21_) comparison for CST and ADS simulations.

**Figure 6 materials-15-05676-f006:**
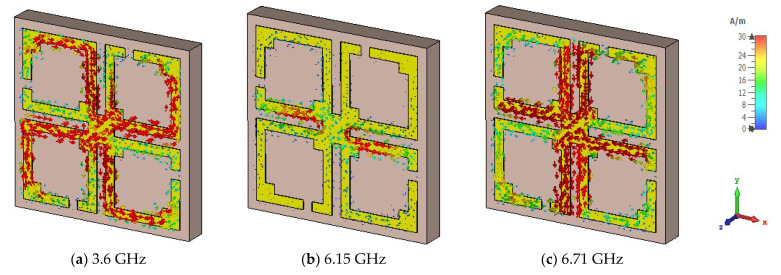
Distributions of the proposed MM surface current at three distinct resonances.

**Figure 7 materials-15-05676-f007:**
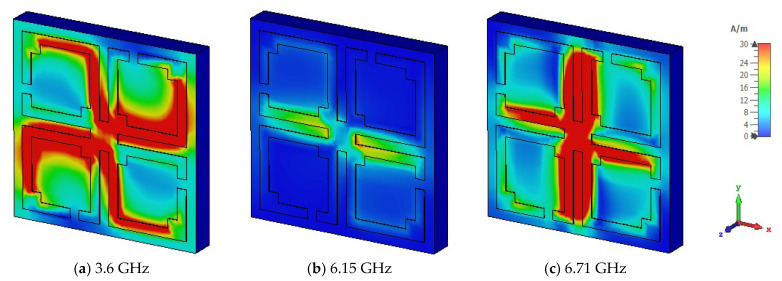
Distributions of the magnetic field at the three distinct resonance frequencies.

**Figure 8 materials-15-05676-f008:**
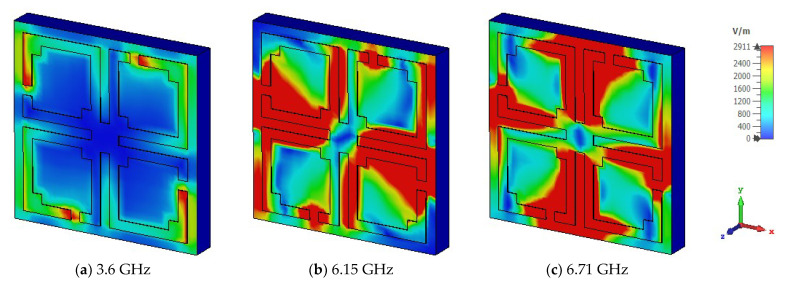
Electric field distribution of MM structure at three different resonance frequencies.

**Figure 9 materials-15-05676-f009:**
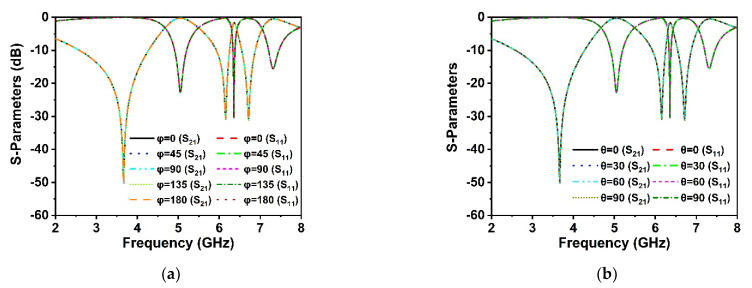
Polarization-independent performance of the proposed unit cell for (**a**) normal incidence and (**b**) oblique incidence.

**Figure 10 materials-15-05676-f010:**
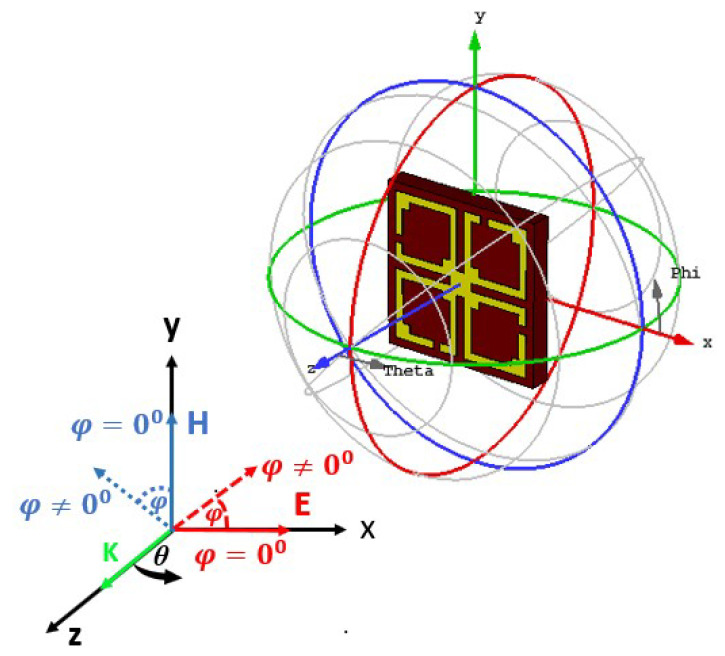
Polarization insensitivity at various angles of incidence.

**Figure 11 materials-15-05676-f011:**
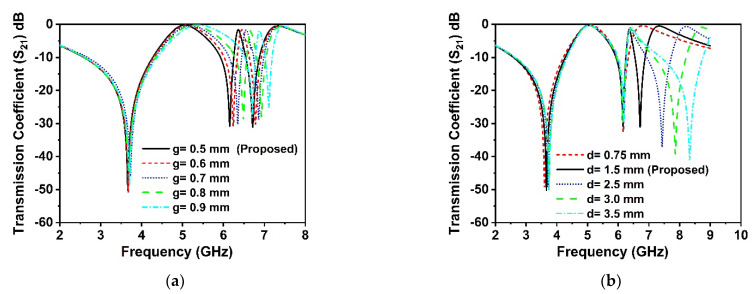
Analysis of transmission coefficients with distinct (**a**) split gap *‘g’* and (**b**) middle connecting strip *‘d’*.

**Figure 12 materials-15-05676-f012:**
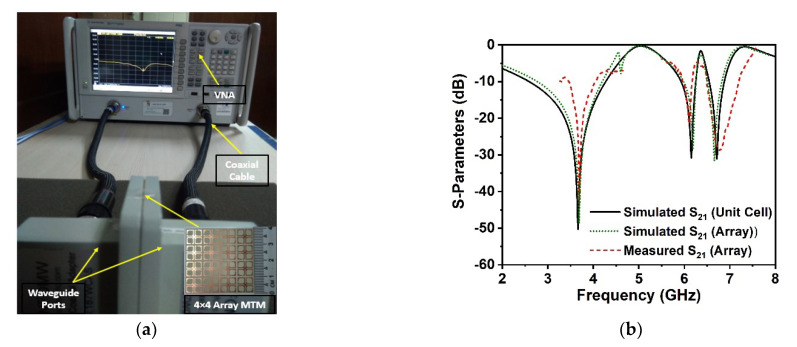
(**a**) Photographs of the array experimental setup with the fabricated prototype (waveguide ports: A-INFOMW, P/N:187WCAS and A-INFOMW, P/N:137WCAS). (**b**) Comparison of the simulated and experimental results of the developed DNG MM structure.

**Figure 13 materials-15-05676-f013:**
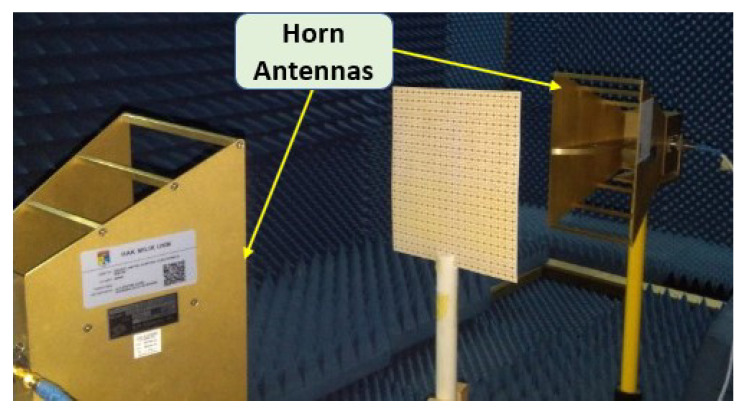
Photograph of the far-field measurement setup.

**Figure 14 materials-15-05676-f014:**
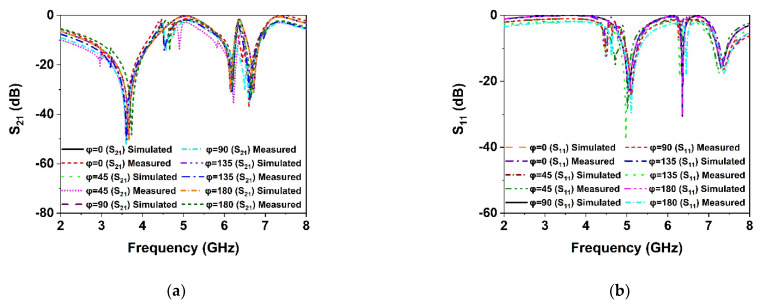
Polarization angle insensitivity experimental results (**a**) *S*_21_ and (**b**) *S*_11_.

**Figure 15 materials-15-05676-f015:**
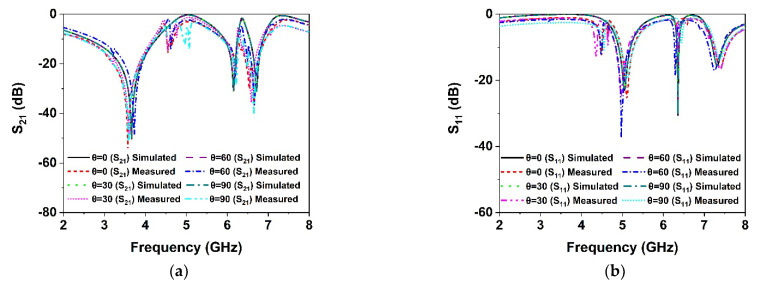
Incident angle insensitivity experimental results (**a**) *S*_21_ and (**b**) *S*_11_.

**Figure 16 materials-15-05676-f016:**
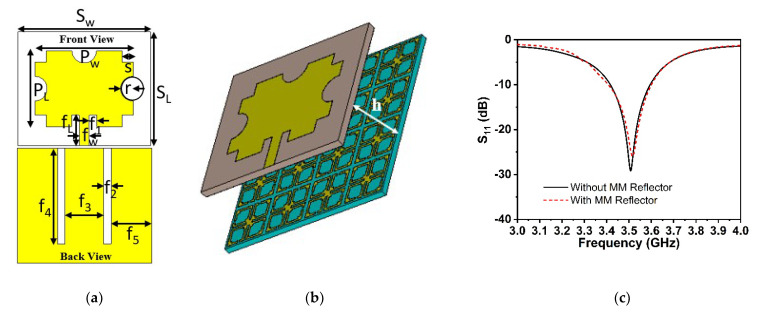
(**a**) Proposed antenna geometry. (**b**) Antenna with a developed MM reflector. (**c**) Reflection coefficient of the antenna with and without MM reflector.

**Figure 17 materials-15-05676-f017:**
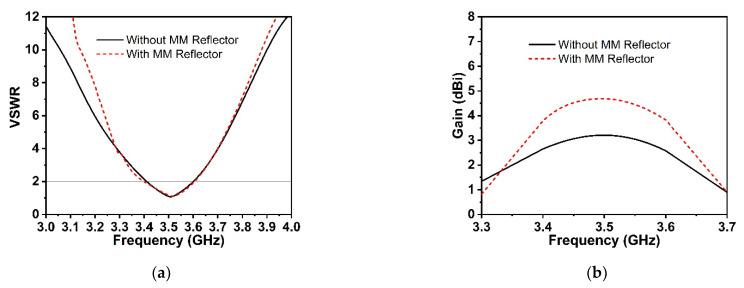
Comparison of the proposed antenna results with and without MM reflector. (**a**) VSWR. (**b**) Gain.

**Figure 18 materials-15-05676-f018:**
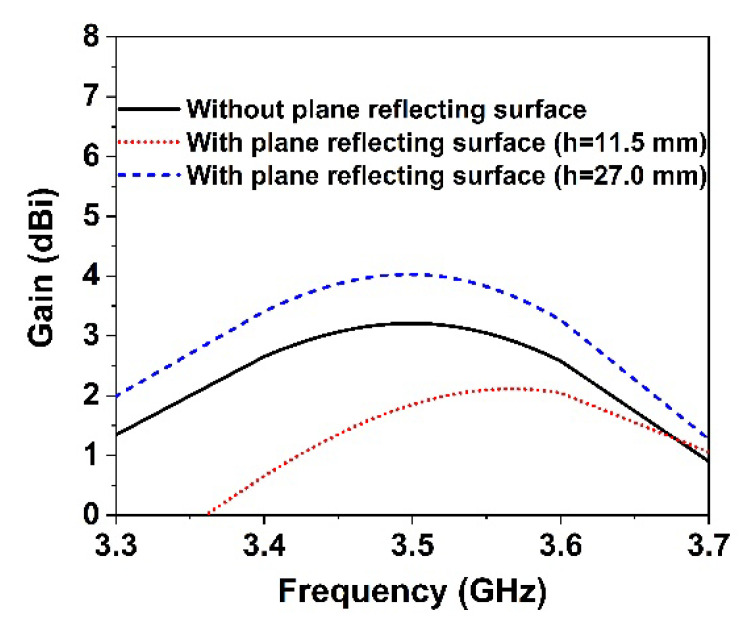
Gain comparison curve of the proposed antenna without and with plane reflecting surface.

**Figure 19 materials-15-05676-f019:**
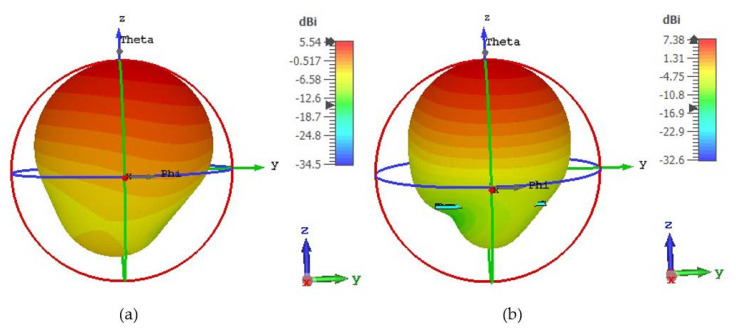
Three-dimensional Far-field results of the proposed antenna (**a**) without MM and (**b**) with MM reflector at 3.5 GHz.

**Figure 20 materials-15-05676-f020:**
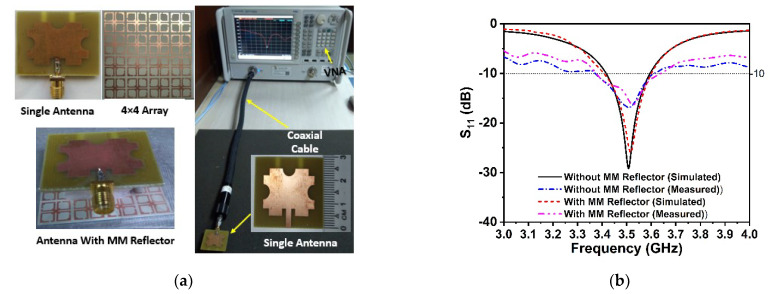
(**a**) VNA experimental arrangement with fabricated prototype. (**b**) Comparison of the *S*_11_ experimental and CST-simulated performance of the proposed antenna with and without the MM reflector.

**Figure 21 materials-15-05676-f021:**
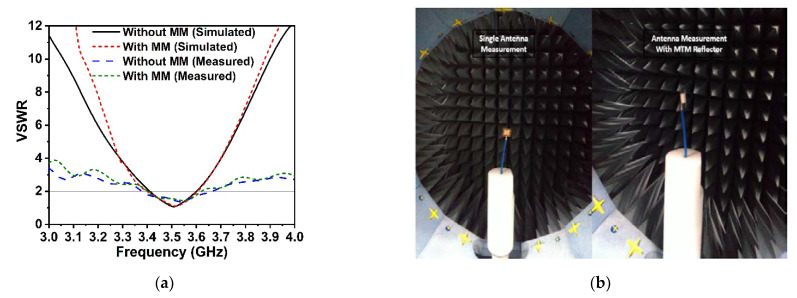
(**a**) Experimental and CST-simulated VSWR values of the proposed antenna with and without MM reflector. (**b**) Photographs demonstrate the antenna efficiency, radiation pattern, and gain measuring procedure at SATIMO near-field system lab, UKM.

**Figure 22 materials-15-05676-f022:**
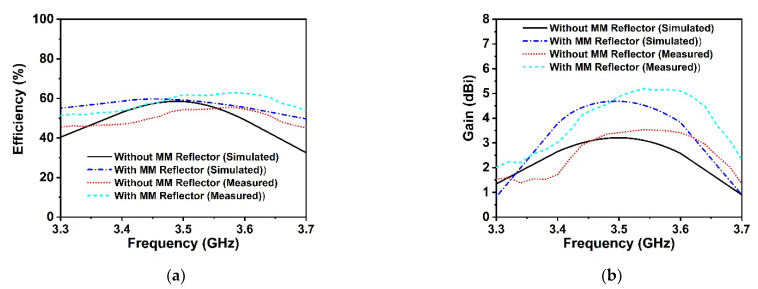
Comparison of (**a**) efficiency and (**b**) gain of the developed antenna with and without MM reflector.

**Figure 23 materials-15-05676-f023:**
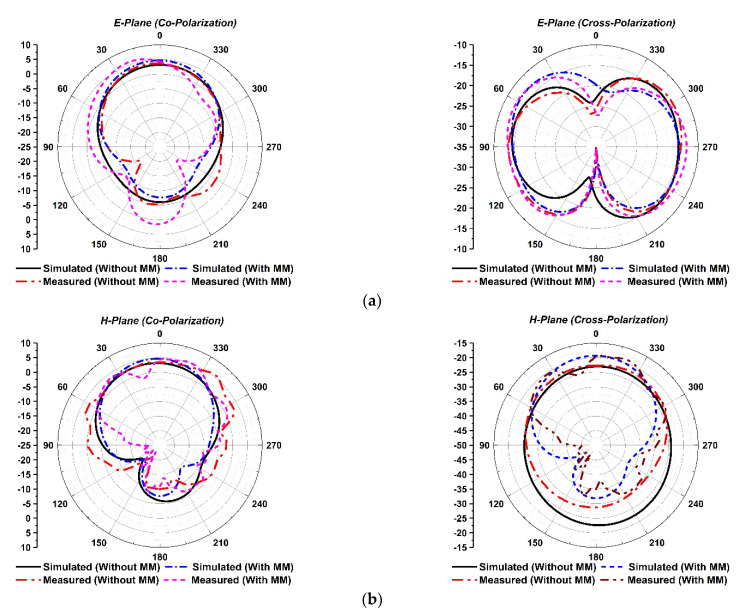
Radiation patterns with and without MM reflector at 3.5 GHz: (**a**) E-plane; (**b**) H-plane.

**Table 1 materials-15-05676-t001:** Parameter values of the proposed DNG MM structure.

Parameter	Size (mm)	Parameter	Size (mm)	Parameter	Size (mm)
*L_s_*	10	*W_s_*	10	*f*_1_, *f*_2_, *f*_3_, *f*_4_, *g*	0.5
*L_r_*	4.25	*W_r_*	4.25	*f*_7_, *d*_1_, *d*_2_	1.5
*f* _5_	2.75	*f* _6_	0.75	*d* _3_	1.0

**Table 2 materials-15-05676-t002:** Effective parameter characteristics of the proposed DNG MM structure.

Parameters	Frequency Range (GHz)	Bandwidth Threshold
*S* _21_	2.5–4.3, 5.9–6.3, 6.5–6.9	$S_{21}<-10\;\mathrm{dB}$
$\varepsilon_{r}$	3.6–5, 6.15–6.34, 6.7–7.24	$\varepsilon_{r}<0$
$\mu_{r}$	3.74–4.39, 6.19–6.26, 6.75–6.96	$\mu_{r}<0$
DNG region	3.74–4.39, 6.19–6.26, 6.75–6.96	$\varepsilon_{r}\;\;\mathrm{and}\;\;\mu_{r}<0$
*n*	3.6–4.9, 6.17–6.35, 6.7–7.1	n<−1.5

## Data Availability

Not applicable.

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
