# Peer review of "Symmetric Engineered High Polarization-Insensitive Double Negative Metamaterial Reflector for Gain and Directivity Enhancement of Sub-6 GHz 5G Antenna"

_materials, 2022, doi:10.3390/ma15165676_

Round 1

Reviewer 1 Report

The paper main idea is interesting, but its presentation must be carefully reviewed. Furthermore, some fundamentals concepts are not clear. For example:

1)If the operation frequency is 3.5 GHz, why so many words and figures were used to describe the results up to 8 GHz?

2)The authors claim that one of the achieved features is the independence of the incidence angle. However, it was not verified experimentally.

3) In order to characterize the proposed MM, the authors used CST software. It is not clear if  in the numerical simulation plane wave ports were employed. If it is the case, how the results  could be compared to the experimental results obtained with TE mode in the waveguide ports?

4) In Fig. 18, despite the implicit difficulty to perform the measurements, the obtained results are not in good agreement with numerical results. They could be improved.

Additional observations are presented below.

-----------------------------------------------------------

In the abstract,

Instead “… The unit cell is designed on a low loss Rogers RO3010 substrate,...”,

 “The MM is designed on a low loss Rogers RO3010 substrate,…” seems to be more appropriate.

In “… gated up to 1800 angles of incidence…”, please, verify the typo (1800). The value (180º) will be discussed later.

-----------------------------------------------------------

Lines 70/71

The authors claim “The objective of this study is to improve the gain and directivity of a 5G antenna by incorporating an MM reflector.”. However, as gain and directivity are related, we propose “… to improve the gain of a 5G antenna…”

-----------------------------------------------------------

Line 115/116

Please, review the text: “… The DNG values have been noticed in the vicinity of  the resonance frequency of S21, where negative permittivity:…”. … “where negative permittivity”? or “with negative permittivity”?

-----------------------------------------------------------

Line 130

Please, review the text “The layout of the manuscript is organized after the introduction;”

-----------------------------------------------------------

Line 135

Please, review the text “…  while section six and seven … “. “… while sections six and seven…”

-----------------------------------------------------------

Line 139/140

Please, review the text: “…The performance comparison table with related recent papers is presented in section ten and summarizes the paper in section eleven.”, “… summarizes the paper in section eleven.” this doesn't seem correct

-----------------------------------------------------------

Line 151

Verify the following statement: “area of 0.12×0.12×0.15”. For which frequency the wavelength is considered? Furthermore, 0.15  does not seem right.

-----------------------------------------------------------

Fig.1/Table 1.

Despite the dimensions were numerically optimized, how were the initial dimensions determined? Please, add some words about it.

 -----------------------------------------------------------

Line 187, equations (1)-(4)

The expressions cited in [33] are only partially presented, making it difficult to understand the adopted procedure. Please, include (2b), X definition, (3a), (3b) (see reference [33]), for a better understanding.

-----------------------------------------------------------

Line 189

“integer number” instead “integernumber”

-----------------------------------------------------------

Page 6, Fig. 3(a).

Please verify. The results presented as permittivity resemble to the impedance.

-----------------------------------------------------------

Page 12,

In Fig. 10, please, verify the x, y and z axis coordinates. they do not obey the right-hand rule.

In Fig. 10, it is not clear where are theta and phi. We strongly suggest to replace the Fig. 10 by the figure generated by the CST software, with a clear indication of the coordinate axis.

-----------------------------------------------------------

Page 13

Fig. 11, vertical axis – If it’s the reflection coefficient, it must be S11. However, it must be the transmission coefficient.

-----------------------------------------------------------

Page 14

Fig. 12 – Please, cite the waveguide ports reference. This is necessary so that it is possible to analyze which operating mode is being considered.

Other question: the same waveguide port pair was used for the all the frequency range, from 2GHz to 8 GHz?

-----------------------------------------------------------

How the independence of the polarization and the incidence angle were verified? They were verified only numerically?

-----------------------------------------------------------

Page 15 and Page 16

In Fig. 13 and Fig. 14, please, replace MMT, by MM, to be in accordance with the text.

-----------------------------------------------------------

In order to obtain a better verification of the proposed models and experimental results, numerical results could be compared to measured results.

-----------------------------------------------------------

Page 17

I am not sure about the contribution of Figs. 16 and 17 for the results analysis. Thus, Figs, 16 and 17 , and respective comments, could be suppressed.

Author Response

Dear Reviewer,

I hereby resubmit our revised manuscript TitledSymmetric Split Ring Resonator Based Double Negative Metamaterial Reflector for 5G Antenna Gain and Directivity Enhancement” of Metamaterial Research Area for possible review and publication in the Material. The revision has been marked in red color in the Final revised manuscript.

Reviewer 2 Report

Dear Authors,

The following are the comments in this regard:

  1. The title should be "Symmetric Engineered High Polarization Insensitive Double Negative Metamaterial Reflector for Gain and Directivity Enhancement of Sub-6 GHz 5G Antenna"
  2. There are numerous typo and grammar errors throughout
  3. The structure offers only a slight enhancement in the gain. Enhancement in gain offered by other antennas should also be compared in Table 3.
  4. Radiations patterns: 2D patterns should be provided for co-polarization and cross-polarization with and without MMs. Also experimental results for the same are needed.
  5. 5G technology uses MIMO. Here a single antenna has been demonstrated. How can we justify it for 5G applications?
  6. How will the structure perform with MIMO concept? 
  7. There are many 5G antennas reported in the literature that offer better gain. How do the authors justify the use of MMs? Can this structure be used with other antennas as well and enhance their gain?

Author Response

Dear Reviewer,

I hereby resubmit our revised manuscript TitledSymmetric Split Ring Resonator Based Double Negative Metamaterial Reflector for 5G Antenna Gain and Directivity Enhancement” of Metamaterial Research Area for possible review and publication in the Materials. The revision has been marked in red color in the Final revised manuscript. Please see the attached file for authors reply.

Reviewer 3 Report

Many researches have already been performed on improving antenna gain and directivity using DNG. The polarization stable characteristics proposed in this paper were presented through simulations rather than experimental proofs, which should be verified through experiments. The authors should show the beam pattern through measurements and comparisons. Also, there is no detailed description of the measurement setup shown in Figure 12.

Author Response

(The authors gave the same response as above.)

Reviewer 4 Report

The manuscript demonstrates a symmetric engineered high polarization insensitive double negative metamaterial reflector with frequency tunable features for fifth generation antenna gain and directivity enhancement. This work is interesting and orgnized well, we recommend its publication in materials.

Some minor concerns are suggested to be solved.

  1. In the abstract ‘The 4×4 array of the MM has been utilized on the backside of the 5G antenna as a reflector, generating additional resonances that contribute to enhancing the antenna gain and directivity.’ So how much enhancing of the antenna gain and directivity? Authors should point out it which is most important information in this paper.
  2. It’s best not to take direct screenshots of sketches in Figure 1.(b).
  3. Why the Measured S21is discontinuities in Fig.12(b)?
  4. The dimensions of the antenna should be given in the paper.
  5. It is sugggested that a table that lists the performance comparision with the state-of-the-art works is presented.
  6. Some relevant references may help enrich the introduction, e.g., Opto-Electronic Advances 3, 200002 (2020), Nano letters, 2017, 17(6): 3752-375, and Science China Physics, Mechanics & Astronomy, 2015, 58(9): 1-18

Author Response

(The authors gave the same response as above.)

Round 2

Reviewer 1 Report

The paper quality has been considerably improved and most of my considerations have been addressed. Thank you  for this revised version.

However, some mandatory questions remain or were introduced in this version:

-----------------

In page 12, Fig. 10, the CST figure is confused. Furthermore, in the incidence angles description only phi angles are depicted. We expected theta angle indication.

-----------------

Line 490, SWR equation is not correct.

SWR=(1+|S11|)/(1-|S11|)

-----------------

Lines 209-212, please verify the misalignment.

-----------------

Please, uniformize the caption of Figures 9 and 10. Use “Figure”, instead “Fig.”

-----------------

Author Response

Dear Reviewer

I hereby resubmit our revised manuscript TitledSymmetric Split Ring Resonator Based Double Negative Metamaterial Reflector for 5G Antenna Gain and Directivity Enhancement” of Metamaterial Research Area for possible review and publication in the Materials as attached file as author response and revised manuscript.

Reviewer 2 Report

Dear Authors,

the Manuscript has been thoroughly revised. I have one last comment 

1. Please include the gain enhancement if a plane reflecting surface is placed below the antenna.

Author Response

(The authors gave the same response as above.)

Reviewer 3 Report

The titel of the submitted paper includes "Polarization Insensitive", but there are no verified results about that using the experiments. In response, the authors do not present the measurement setup except the waveguide measurent that does not show the angle and polarization sensitivity.

In case of beam pattern, the authors presented the normalized beam pattern that could not explain the accuracy of the simulation and measurements.

Author Response

(The authors gave the same response as above.)
